# Getting higher on rugged landscapes: Inversion mutations open access to fitter adaptive peaks in NK fitness landscapes

**Leonardo Trujillo**⬤*, **Paul Banse, Guillaume Beslon**⬤

Université de Lyon, INSA-Lyon, INRIA, CNRS, Université Claude Bernard Lyon 1, ECL, Université Lumière Lyon 2, LIRIS UMR5205, Lyon, France

* leonardo.trujillo@inria.fr

**Data Availability Statement:** The source code of all models and stimulation files used in the present

## Abstract

Molecular evolution is often conceptualised as adaptive walks on rugged fitness landscapes, driven by mutations and constrained by incremental fitness selection. It is well known that epistasis shapes the ruggedness of the landscape's surface, outlining their topography (with high-fitness peaks separated by valleys of lower fitness genotypes). However, within the strong selection weak mutation (SSWM) limit, once an adaptive walk reaches a local peak, natural selection restricts passage through downstream paths and hampers any possibility of reaching higher fitness values. Here, in addition to the widely used point mutations, we introduce a minimal model of sequence inversions to simulate adaptive walks. We use the well known NK model to instantiate rugged landscapes. We show that adaptive walks can reach higher fitness values through inversion mutations, which, compared to point mutations, allows the evolutionary process to escape local fitness peaks. To elucidate the effects of this chromosomal rearrangement, we use a graph-theoretical representation of accessible mutants and show how new evolutionary paths are uncovered. The present model suggests a simple mechanistic rationale to analyse escapes from local fitness peaks in molecular evolution driven by (intragenic) structural inversions and reveals some consequences of the limits of point mutations for simulations of molecular evolution.

## Author summary

Ninety years ago, Wright translated Darwin's core idea of survival of the fittest into rugged landscapes—a highly influential metaphor—with peaks representing high values of fitness separated by valleys of lower fitness. In this picture, once a population has reached a local peak, the adaptive dynamics may stall as further adaptation requires crossing a valley. At the DNA level, adaptation is often modelled as a space of genotypes that is explored through point mutations. Therefore, once a local peak is reached, any genotype fitter than that of the peak will be away from the neighbourhood of genotypes accessible through point mutations. Here we present a simple computational model for inversion mutations, one of the most frequent structural variations, and show that adaptive processes in rugged

paper can be found in https://gitlab.inria.fr/letrujil/getting-higher.

**Funding:** L.T. was partially supported by Institut National des Sciences Appliquèes (INSA) Visiting Professor Fellowship. P.B. was supported by Ministère de l'Enseignement Supérieur et de la Recherche. The funders had no role in study design, data collection and analysis, decision to publish, or preparation of the manuscript.

**Competing interests:** The authors have declared that no competing interests exist.

landscapes can escape from local peaks through intragenic inversion mutations. This new escape mechanism reveals the innovative role of inversions at the DNA level and provides a step towards more realistic models of adaptive dynamics, beyond the dominance of point mutations in theories of molecular evolution.

## Introduction

The fitness landscape is a very influential metaphor introduced by Wright [1] to describe evolution as explorations through a "field of possible genes combinations", where high values of fitness are represented as peaks separated by valleys of lower fitness. The topography of the fitness landscape has important evolutionary consequences, e.g. speciation via reproductive isolation [2]. Within this framework, the evolution of any population can be conceptualised as adaptive walks driven by successive mutations constrained by incremental or neutral fitness steps. Thus, in the absence of additional evolutionary forces such as drift or environmental variations, once a population reaches a local peak, natural selection hampers any further mutational paths that decrease fitness. However, there are empirical evidences showing that populations do not stop indefinitely at a local peak and can explore alternative trajectories on the landscape [3–7], *ergo*, the following question arises: *How does evolution escape from a local peak to a fitter one?*

Since it has been formulated, considerable progress have been made on this question [8–22]. However, conventional theoretical approaches still state that a genotype mutates into another through *point mutations* (e.g. single-nucleotide variations). If one takes a look at the molecular scale of DNA and the different mutation types, this may seem contradictory as it is well known that many other kinds of variation operators (including insertions, deletions, duplications, translocations and inversions) act on the genome. Hence, a fundamental aspect of this challenge is to understand—at the scale of molecular evolution—the roles played by these different mutation types. Indeed, there is a gap between the theoretical models, that account for a very limited set of mutations types—typically only point mutations—and the reality of molecular evolution, where multiple variation operators act on the sequence.

As a contribution to bridge this gap, we present a minimal DNA-inspired mechanistic model for inversion mutations, and explore their relationship with the escape dynamics from local fitness peaks. Inversion mutations are one of the most frequent chromosomal rearrangements [23, Ch. 17.2] with lengths covering a wide range of sizes. For example, in Long Term Evolution Experiments with *E. coli*, chromosomal rearrangements have been characterised by optical mapping (hence limiting the resolution to rearrangements larger than 5000 bp) [24]. In this study, 75% of evolved populations showed inversion events—ranging in size from $\sim 164$ Kb to $\sim 1.8$ Mb [24] (for other examples of large inversions in different clades see [25]). With the development of novel sequencing technologies [26–28], it has been possible to identify intragenic (submicroscopic) DNA inversions, for example, an inversion of seven nucleotides in mitochondrial DNA, resulting in the alteration of three amino acids and associated to an unusual mitochondrial disorder [29]. Intragenic inversions have also been suggested to be an important mechanism implied in the evolution of eukaryotic cells [30]. Although chromosomal inversions are ubiquitous in many evolutionary processes [23–27, 29–38], very little is known about their theoretical description and computational simulation at the sequence level, as models generally focus on very large inversions (typically larger than a single gene), hence on their effect on synteny (or the deleterious effects at breakpoints), but neglect the possibility that small inversions occur inside coding sequences [39].

Here, we simulate a representation of molecular evolution of digital organisms (replicators), each of which contains a single piece of DNA. We engineer a computational method to cartoon the double-stranded structure of DNA, and simulate inversion-like mutations consisting of a permutation of a segment of the complementary strand, which is then exchanged with the main strand segment (see Methods, schema 6). For the sake of simplicity, we consider digital genotypes made up of binary nucleotides (i.e. a binary alphabet {0, 1} instead of the four-nucleotides alphabet {A, T, C, G}). The sequences are arranged in circular strings with constant number of base-pairs. In an abstract sense, the model mimics the molecular evolution of some viruses [40] and (animal) mitochondrial DNA [41] with compact genomes and closed double-stranded DNA circles [40, 42, 43]. It is very important to emphasise that our computational model simulates intragenic-like mutations [29, 30, 44]. We are modelling asexual replication, therefore recombination is not considered. To build rugged fitness landscapes, we adopt the well known Kauffman NK model, where N denotes the length of the genome and K parameterizes the "epistatic" coupling between nucleotides [45–48]. We do not include environmental changes, so the landscape remains constant through the simulations. Finally, it is worth mentioning that all our simulations were conducted in the evolutionary regime of strong selection weak mutations (SSWM) [49, Ch. 5].

## Results

### Who is next to whom: The mutational network

We first study how inversion mutations can increase the number of accessible mutants. For this we translate the canonical notion of neighbour genotypes (see Methods, Eq 1) into a graph theory approach, and analyse the simplest and most familiar geometric object in molecular evolutionary theory: the discrete space of binary sequences (unless explicitly stated we will henceforth consider only binary alphabets). Thinking topologically, all the sequences $\mathbf{x}$ with $N$ binary-nucleotides $x_i \in \{0, 1\}$, $\forall i = 1, \ldots, N$ and $N \in \mathbb{N}_{\geq 2}$, define the set $\mathcal{X} \in \{0, 1\}^N$ of $2^N$ possible genotype combinations. A canonical measure to characterise the topology of the set $\mathcal{X}$ is the Hamming distance

$$d_H(\mathbf{x}, \mathbf{x}') \coloneqq \sum_{i=1}^{N} (x_i - x_i')^2.$$

A convenient way to organise such a set is by graphs connecting two sequences $\mathbf{x}$ and $\mathbf{x}'$ that differ by one point mutation (i.e. $d_H(\mathbf{x}, \mathbf{x}') = 1$). This is the so-called Hamming graph $\mathcal{H}(N, 2)$—a special case of the hypercube graph $\mathcal{Q}_N$ (the well-known graph representation of the genotype space). On the other hand, the Hamming distance for inversion mutations forms a set of integers satisfying

$$0 \leq d_H(\mathbf{x}, \mathbf{x}') \leq N,$$

meaning that, contrary to point mutations, the Hamming distance of inversion mutations range from zero to $N$ (see Methods). Note that if the inversion spans the entire chromosome, then $d_H(\mathbf{x}, \mathbf{x}') = N$, all loci have changed, but it also implies that 5'— 3' becomes 3'—5' and vice versa and nothing has changed biologically.

We propose that, for a sequence $\mathbf{x} \in \mathcal{X}$, the mutation operation (i.e. the mechanistic representation of point or inversion mutations) build the set $\mathcal{N}_v(\mathbf{x})$ of accessible mutants $\mathbf{x}' \in \mathcal{N}_v(\mathbf{x})$, $\forall \mathbf{x}' \neq \mathbf{x}$, $\Rightarrow d_H(\mathbf{x}, \mathbf{x}') \neq 0$ (the subindex $v$ denotes the type of mutation: $P$ for point mutations and $I$ for inversion mutations). Therefore, the number of neighbouring mutants

can be reformulated as

$$D_v(\mathbf{x}) = |\mathcal{N}_v(\mathbf{x})|.$$

Inversion's combinatorics is not trivial since it involves the permutation of a subsequence and its flips between each strand (see Methods, schemata 2 and 6). Nevertheless, from the algorithmic point of view, for a given genotype **x** the mutational operations can be used to enumerate all the accessible mutants (see Methods, Algorithm 1: `Mutate`). Also, the combinatorics can be represented as a directed multigraph of mutations $m(\mathbf{x}), \forall \mathbf{x} \in \mathcal{X}$ (see the mathematical definition in [50, p.8]), i.e. the ordered triple

$$m = (V(m), E(m), I_m),$$

where $V(m) = \mathbf{x} \cup \mathcal{N}_v(\mathbf{x})$ is the set of vertices (formed by a given genotype **x** and its mutated genotypes $\{\mathbf{x}'\} \in \mathcal{X}$), $E(m)$ is the set of directed edges (from genotype **x** to a mutated genotype $\mathbf{x}'$) and $I_m : \mathcal{X} \rightarrow \mathcal{X}$ is an incidence relation that associates to each element of $E(m)$ an ordered pair of $V(m)$. In fact, the incidence relation $I_m$ corresponds to a mutation operation (see Methods, Eqs (3), (4) and (5)). As an example, in Fig 1 we display the atlas of accessible-mutants for $N = 4$, constructed by calculating all inversion mutations for each one of the $2^4$ (wild-type) sequences (central red dots). In this example (see also Table 1), it is verified that the number of accessible-mutants for inversion mutations $D_I(\mathbf{x}), \forall \mathbf{x} \in \{0, 1\}^4$ is in the set {7, 8, 13} (unlike the case for point mutations, which must be a singleton, i.e. the single-value set: $D_P(\mathbf{x}) \in \{4\}, \forall \mathbf{x} \in \{0, 1\}^4$). We can also verify that $\min(D_I(\mathbf{x})) \geq \max(D_P(\mathbf{x}))$. In Table 1 we show the enumeration of accessible-mutants for inversions and point mutations for genome sizes ranging from $N = 2$ to 10. From Fig 1 and Table 1, we can see that the combinatorics of the inversion mutations is not trivial. We can verify that the maximum number of accessible mutants is equal to $N^2 - N + 1$, which corresponds to the trivial cases of genotypes **x** with $x_i = 0, \forall i \in \{0, \ldots N\}$ and $x_i = 1, \forall i \in \{0, \ldots N\}$. Note that for a circular sequence of size $N$, the total number of inversion mutations is $N^2$, while for point mutations this number is equal to $N$. However, the number of mutants accessible by inversions is lower than the total number of inversions mutations ($D_I < N^2$). This is due to "degenerate" inversion mutations: several inversions—occurring between different loci and/or for different interval sizes—may mutate the initial sequence to the same accessible mutant (see the multiple edges in Fig 1). In Fig 1, we can also verify that there are loops (an "edge" joining a vertex to itself), that is, "invariant inversions" that preserves the nucleotide sequence after the inversion operation (i.e. $d_H(\mathbf{x}, \mathbf{x}') = 0$). It can easily be shown that the fraction of invariant inversions converges to $1/N$ (see S1 Text, section 1). A very important consequence of inversions is that mutated sequences can differ with the wild type by more than one nucleotide, i.e. $d_H(\mathbf{x}, \mathbf{x}') > 1$ (edges colours in Fig 1 denote the values of the Hamming distance). This result allows us to gain a first insight of how inversions can promote the escape from local fitness peaks: they can "connect", in a single mutational event, genotypes that are at two or more point-mutational steps away. It is pertinent to remark that the combinatorics of inversions for alphabets with size $|\mathcal{A}_\mathcal{L}| = 2n, \forall n \geq 2$ would imply even more connections.

It should be noted that the inversion combinatorics would be slightly different for linear sequences. In that case, the number of possible inversions is $N(N + 1)/2$.

Up to this point, we have shown how inversion mutations can actually broaden the horizon of evolutionary exploration in the genotype space.

**Inversions rewire the adjacency of the genotype space.** Now, for each directed graph of mutations $m(\mathbf{x})$, we can associate a graph $M(\mathbf{x})$ on the same set of vertices $V(m)$. Corresponding to each directed edge of m, there is an edge of M with the same ends (loops being

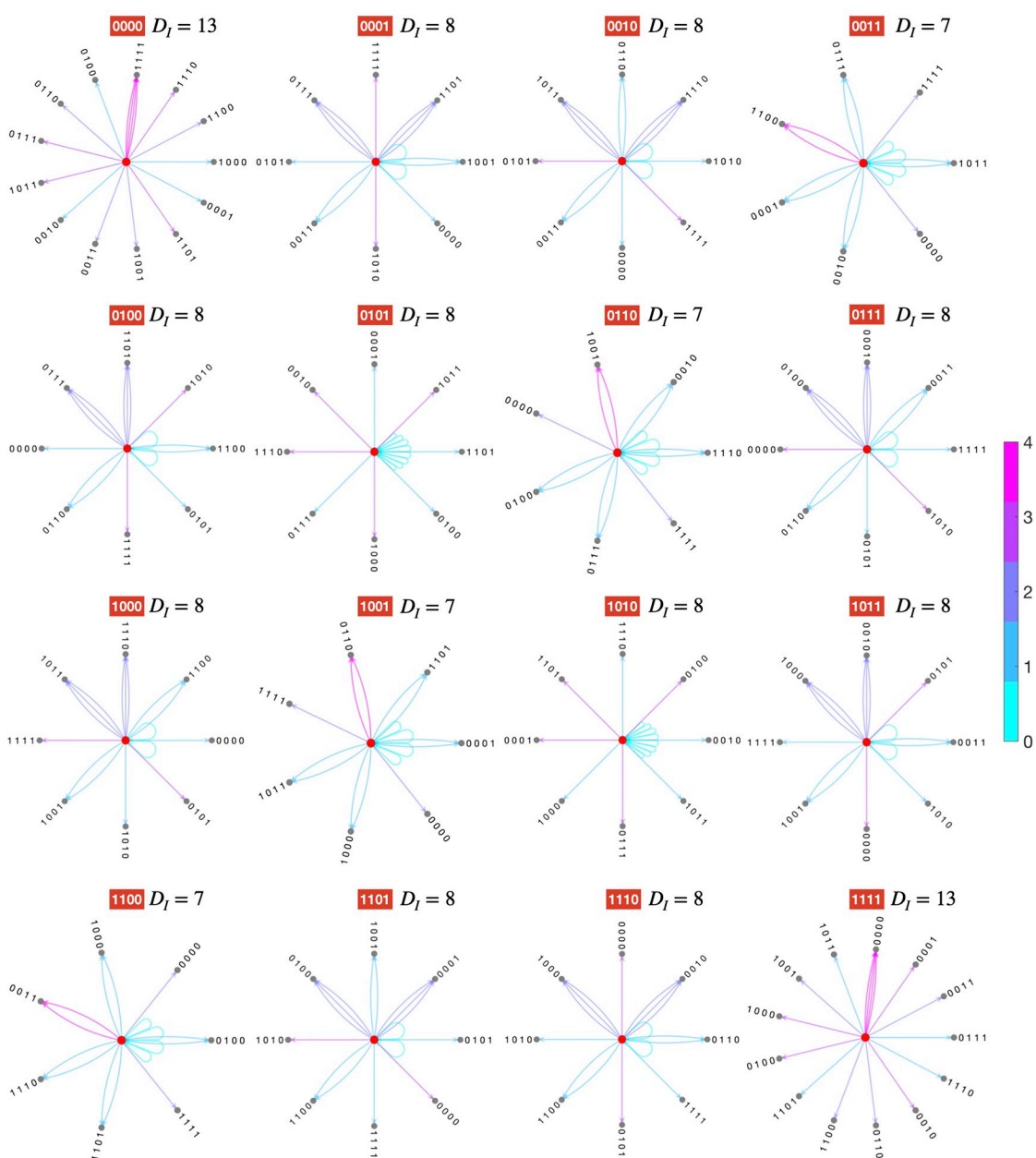

**Fig 1. Atlas of accessible-mutants.** Example of the total enumeration of inversion mutations, represented as *graphs of accessible mutants*, for each one of the $2^4$ genotypes (central red nodes) with size $N = 4$. Edges colour quantifies the Hamming distance $d_H(\mathbf{x}, \mathbf{x}')$, between the central nodes $\mathbf{x}$ (wild-types) and their mutants $\mathbf{x}'$. Each wild type is labeled in red and its number of accessible mutants $D_I$ is also displayed. Let us remark that this enumeration also depends on the fact that in our model the sequences are circular (i.e. periodic boundary conditions: $x_{N+i} = x_i$, $\forall i \in \{1, \ldots, N\}$).

excluded). In this sense, the graph M($\mathbf{x}$) is the underlying (simple) graph of the directed graph m($\mathbf{x}$). That is, the graphs without self- and directed-edges so that when several edges (mutations) connect the genotype $\mathbf{x}$ with the same accessible-mutant $\mathbf{x}'$, only one (undirected) edge is kept. Thus, every directed graph m($\mathbf{x}$) defines a unique, up to isomorphism, reduced graph M($\mathbf{x}$) (see the mathematical definition in [50], p.3). Now it is natural to do the union of each graph M($\mathbf{x}$), to describe how genotypes can be reached from somewhere in the genotype space

**Table 1. Enumeration of accessible-mutants.** Number of neighboring mutants accessible via inversion mutations $D_I$ and via point mutations $D_P$ for all genomes of size $N \in [\![2, 10]\!]$ (subscripts numbers denote the number of occurrence of each value of $D_I$ and $D_P$).

| Genome size $N$ | Number of accessible-mutants via inversions $D_I$ | Number of accessible-mutants via point mutations $D_P$ |
|---|---|---|
| 2 | $2_2$, $3_2$ | $2_4$ |
| 3 | $5_6$, $7_2$ | $3_8$ |
| 4 | $7_4$, $8_{10}$, $13_2$ (See Fig 1) | $4_{16}$ |
| 5 | $13_{20}$, $17_{10}$, $21_2$ | $5_{32}$ |
| 6 | $16_{30}$, $17_{18}$, $18_2$, $22_{12}$, $31_2$ | $6_{64}$ |
| 7 | $25_{70}$, $29_{42}$, $37_{14}$, $43_2$ | $7_{128}$ |
| 8 | $28_{16}$, $29_{52}$, $30_{112}$, $32_2$, $34_8$, $36_{48}$, $46_{16}$, $57_2$ | $8_{256}$ |
| 9 | $39_6$, $40_{18}$, $41_{234}$, $45_{162}$, $52_{36}$, $53_{36}$, $64_{18}$, $73_2$ | $9_{512}$ |
| 10 | $45_{100}$, $46_{150}$, $47_{420}$, $50_2$, $52_{40}$, $53_{200}$, $62_{40}$, $63_{50}$, $77_{20}$, $91_2$ | $10_{1024}$ |

in one mutation operation. We call this object the mutational network. It is defined as:

$$\mathcal{M}(\mathcal{X}) := \bigcup_{\mathbf{x} \in \mathcal{X}} M(\mathbf{x}).$$

For point mutations the mutational network is the Hamming graph $\mathcal{H}(N, 2)$ [51, p. 230] (as we can see in Fig 2A, 2B and 2C for $\mathcal{H}(4, 2)$, $\mathcal{H}(7, 2)$ and $\mathcal{H}(10, 2)$ respectively), which is isomorphic to the canonical genotype space $\mathcal{H}(N, 2) = \mathcal{Q}_N$. The notion of isomorphism means that the mutational network for point mutations preserves the adjacency of the edge structure of the genotype space. Historically, the canonical graph of the genotype space overshadowed the richness of the (full) mutation graph, since theoretically only point mutations are usually considered as generators of mutational networks. For inversions, the mutational network does not necessarily inherit the (local) topology of the genotype space. For example, Fig 2D, 2E and 2F outline the structure of the mutational networks for inversion mutations for $N = 4$, 7 and 10. From the point of view of graph theory, inversion mutations "rewire" the adjacency of the genotype space, i.e. they link genomes such that $d_H(\mathbf{x}, \mathbf{x}') \geq 1$. Also, in graph terms, the total number of accessible mutants per genotype corresponds to the node's degree $\kappa_{\mathbf{x}}$ (defined as the number of edges in the graph incident on $\mathbf{x}$ [50, p. 3]). Therefore, $\kappa_{\mathbf{x}} = D_v(\mathbf{x})$. On the other hand, the average node degree quantifies accessible-mutants (nodes) interconnections:

$$\langle \kappa \rangle := \frac{1}{2^N} \sum_{\mathbf{x}} \kappa_{\mathbf{x}}.$$

It can be verified that for point mutations $\langle \kappa \rangle = N$, while for inversions mutations $\langle \kappa \rangle > N$ (for $N \geq 2$) and therefore the genotypes are "more connected" to each other. Paraphrasing in terms of evolutionary biology, they are "more mutable". In this sense, $\langle \kappa \rangle$ defines a mean mutability, which quantifies the ability to reach a different genome when the sequence undergoes a mutation. This property also holds for linear chromosomes, although as mentioned above, the average of node degrees is smaller since the number of possible inversions is lower than for circular chromosomes.

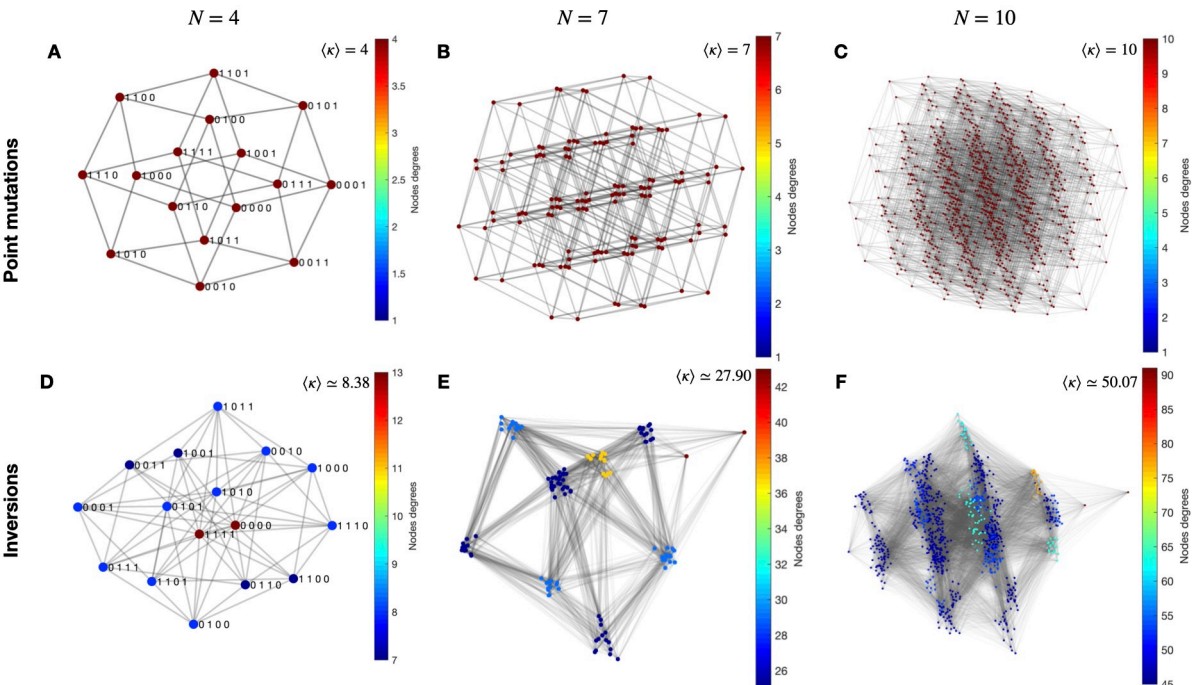

**Fig 2. Mutational networks.** Representative examples for $N = 4$, 7 and 10. Colour indicates the node's degree $\kappa$. The reported values correspond to average node degrees $\langle \kappa \rangle$. The upper graphs show the point mutation case, verifying that the mutational networks are Hamming graphs and therefore isomorphic to their genotype spaces: (A) $\mathcal{H}(4, 2)$; (B) $\mathcal{H}(7, 2)$ and (C) $\mathcal{H}(10, 2)$. The lower graphs (D), (E) and (F) correspond to the inversion mutations cases, where we can note that these mutational networks are not isomorphic to their genotype spaces.

## Inversions can reveal new evolutionary paths

Even though the nature of the genotype-to-fitness function is still largely unknown, an easy way to introduce it into computational models is by assuming that for genotypes $\mathbf{x} \in \mathcal{X}$ there exists a map from the set $\mathcal{X}$ to the real numbers $f : \mathcal{X} \rightarrow \mathbb{R}$. In the graph-based representation, each node (genotype) then possesses a fitness value $f(\mathbf{x})$. This fitness landscape graph F is isomorphic to the hypercube graph $\mathcal{Q}_N$ (i.e. the genotype space) and therefore can also be represented as Hamming graphs, providing a fitness value per node. So, the fitness landscape graph is univocally defined as:

$$\text{F}(\mathcal{X}, \mathcal{N}_P, f) := (\mathcal{H}(N, 2), f).$$

Likewise, as the mutational network we can also define the fitness network $\mathcal{F}$, but in this case the edges are directed from genotypes with lower fitness to genotypes with higher fitness:

$$\mathcal{F}(\mathcal{X}, \mathcal{N}_v, f) := (\mathcal{X}, \ \{(\mathbf{x}, \mathbf{x}') \in \mathcal{X}^2 \mid \mathbf{x}' \in \mathcal{N}_v(\mathbf{x}) \text{ and } f(\mathbf{x}') > f(\mathbf{x})\}, f),$$

which also depends on the neighbouring $\mathcal{N}_v$, with $v$ denoting the type of mutation (P and I for point and inversion mutations respectively). Therefore, the fitness network is the anisotropic version of the mutational network, the direction of the evolutionary paths being fitness-dependent. Precisely what mutational and fitness networks reveal to us is the ensemble of possible evolutionary paths. But in this case, fitness networks are diagrams showing the paths upward and their "altitudes" (fitness values).

To illustrate the definition of fitness network used in this work, we need to build fitness landscapes instances. For that, we use the well-known NK-model [45, 46, 48], recalling that for

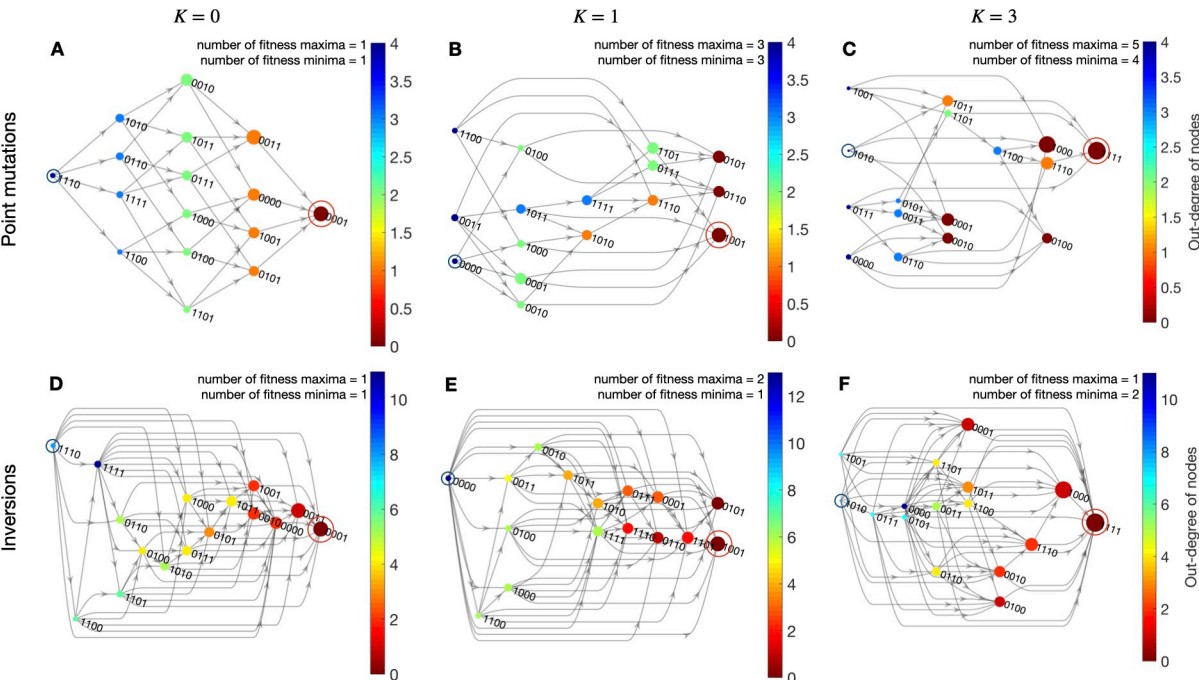

**Fig 3. NK fitness networks for epistatic interactions with random neighbouring.** Representative instances of the NK model for $N = 4$ and their fitness networks in layered representation. The layers are constructed such that each node is assigned to the first possible layer, with the constraint that all its predecessors must be in earlier layers. The colors of the nodes correspond to the values of the out-degrees, i.e. the number of edges going out of a node (note that color scales differ in range between panels). Therefore, nodes with node out-degrees equal to zero correspond to local fitness maxima (sink nodes). The landscapes' ruggedness are: single peaks $K = 0$, intermediate ruggedness $K = 1$ and full rugged case $K = 3$. Node sizes are scaled with fitness values (the best fitness, the largest, and vice versa). Global maximum of fitness are encircled in red. While the global minimum in blue. The total number of fitness maxima and minima are also reported. See S1 Fig for epistatic interactions with adjacent neighbouring.

a genome of size $N$, the parameter $K \in \{0, \ldots, N-1\}$ corresponds to the epistatic coupling between loci and thus tunes the ruggedness of the landscape (for a brief introduction see Methods). Here, we use two neighbourhood coupling models between loci: i) the adjacent model in which the $K$ loci are those closest to a focal locus $x_i$; ii) the random neighbourhood model, where the $K$ loci are chosen randomly among the $N-1$ loci other than $x_i$ (an illustration of epistatic interactions is sketched in NK model in Methods). In Fig 3 we show representative examples of fitness networks engineered for $N = 4$ with $K$ epistatic random neighbours (see S1 Fig for the fitness networks with $K$ epistatic adjacent neighbours). The landscapes range from single peaked $K = 0$ (no epistatic interaction) to full rugged landscapes $K = 3$ (highly connected epistatic interactions). Global fitness maxima and minima are highlighted by encircled nodes. In Fig 3A, 3B and 3C we show an instance of fitness networks for genotypes connected through pathways with point mutation steps. They have the same topology as the genotype space $\mathcal{Q}_{N=4}$ and, therefore, are isomorphic to the graph representation of the fitness landscape $\mathcal{F} \cong F = (\mathcal{H}(4, 2), f)$. When $K = 0$ all the trajectories arrive to the single global maximum of fitness. When we construct the fitness network with inversion mutations, we can verify in Fig 3D, 3E and 3F that there are more paths between all the genotypes, and so it is easier for an evolutionary process to explore more domains of the fitness landscape compared to point mutations. Many of these paths connect genotypes such that $d_H(\mathbf{x}, \mathbf{x}') > 1$, and therefore, are like jumps between distant domains of the landscape. We can also verify that, for a given fitness function $f$, a node that is a local optimum on the fitness graph $F = (\mathcal{H}(N, 2), f)$, is not

necessarily a local peak for inversion mutations in the fitness network. In most of the cases, the fitness landscape is "smoothed out" by inversion mutations, since the notion of local peak fades in the fitness network. However, the local peaks are not always smoothed out, as we can see in Fig 3E for $K = 1$, where genotype 0101 remains as a local peak. This is because 0101 cannot be mutated to genotype 1001 by any inversion mutation (see also the combinatorics of accessible mutants for 0101 in Fig 1). Note that for inversion mutations, it is verified that in some cases the global maximum can be reached from the global minimum in a single evolutionary step with $d_H(\mathbf{x}, \mathbf{x}') \neq 1$, e.g. Fig 3E with $d_H = 2$.

Finally, contrary to point mutations, inversions are not commutative: in many cases, two overlapping inversions applied to a same initial sequence in direct or reversed order lead to different final sequences. This can easily be shown on an example:

$$\begin{array}{ccccc} 0111100 & \xrightarrow{inv(2,3)} & 01\,00\,100 & \xrightarrow{inv(2,4)} & 01\,011\,00 \\ 1000011 & \circlearrowleft & 10\,11\,011 & \circlearrowleft & 10\,100\,11 \end{array}$$

$$\begin{array}{ccccc} 0111100 & \xrightarrow{inv(2,4)} & 01\,000\,00 & \xrightarrow{inv(2,3)} & 01\,11\,0\,00 \\ 1000011 & \circlearrowleft & 10\,111\,11 & \circlearrowleft & 10\,00\,1\,11 \end{array}$$

where, starting from the same sequence, the two inversions ($inv(2, 3)$ and $inv(2, 4)$) give different outcomes depending on their order. Note that, given this property, the classical definition of mutational epistasis does not hold for inversion mutations.

## Getting higher on rugged landscapes

Up to now, we have shown results on the combinatorial (topological) differences between point and inversion mutations. Inversions cannot be mapped to the classical "fitness landscape" metaphor—being better represented through mutational networks and their juxtaposition with fitness landscapes through fitness networks. This is because, for inversion mutations, there are shortcut routes connecting distant sequences (differing by more than one base) in the genotype space and consequently in the fitness landscape. Therefore, this can be interpreted as "escape routes" from local peaks. We want to verify if as a consequence of these escape routes, an evolutionary process will be able to reach higher peaks of fitness. For that, we performed computer simulations in the SSWM setting, where adaptation occurs by sequential fixing of novel beneficial mutations (see Adaptive walks in Methods).

We focus our study on a series of $n$-repetitions of adaptive walks, where the evolutionary process is driven by (random) mutational steps. For a given set of independent initial random genomes with size $N = 100$, $\{\mathbf{x}_0\} \in \{0, 1\}^{100}$, we create two pools of $n = 100$ simulations for point mutations and inversions respectively. As before, we use the NK model to engineer rugged fitness landscapes. In each round, the landscape is the same for simulations with point mutations and inversions, respectively. For independent explorations over (sub)domains of the landscape, we monitor the time-evolution of the fitness values until a fitness optimum is reached. This is when it is verified, in the simulation, that a genotype $\mathbf{x}_v^{\mathrm{loc}} \in \mathcal{X}$ satisfies

$$f(\mathbf{x}') < f(\mathbf{x}_v^{\mathrm{loc}}), \ \forall \mathbf{x}' \in \mathcal{N}_v(\mathbf{x}_v^{\mathrm{loc}}).$$

Subindex $v$ denotes the type of mutation (P for point mutations and I for inversion mutations). Then, we calculate the mean fitness value per $K$ as:

$$\langle f_v \rangle_K := \frac{1}{n} \sum_{i=1}^{n} f(\mathbf{x}_v^{\mathrm{loc}}(i))\bigg|_K,$$

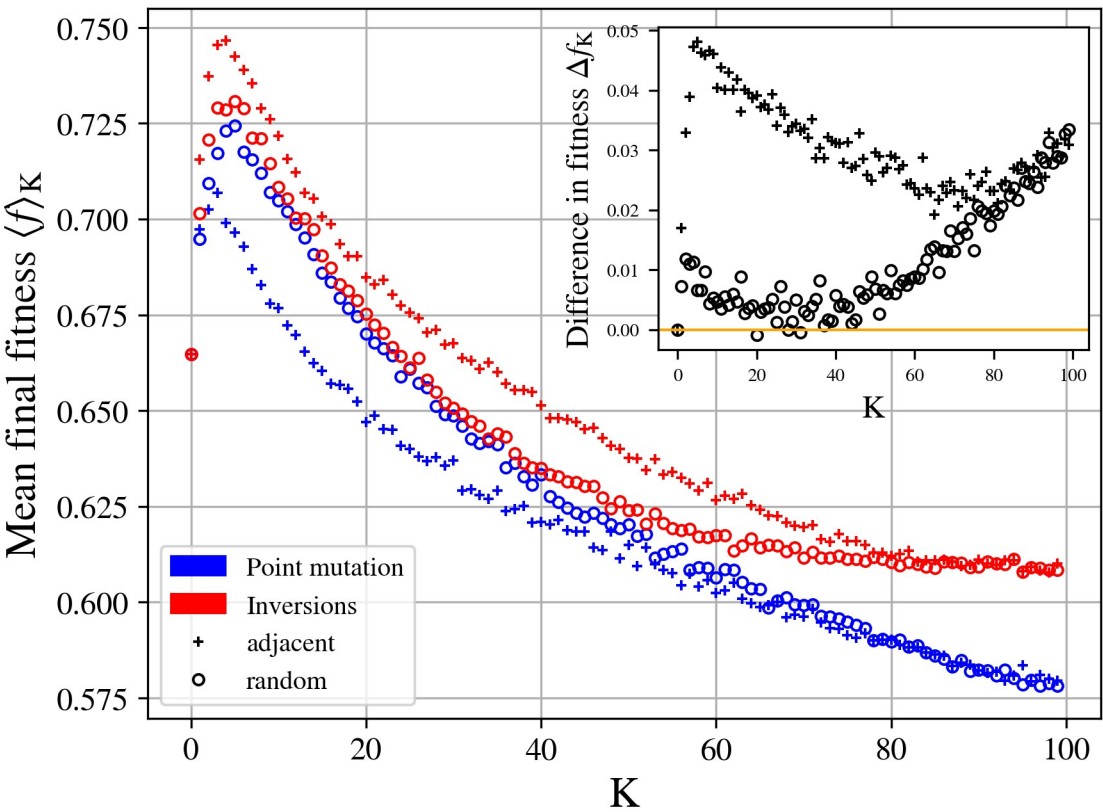

**Fig 4. The average value of local fitness maxima suggests the escape from local fitness peaks by inversion mutations.** Changes in mean final fitness for different epistatic parameter K for inversion (red) and point mutations (blue), averaged for 100 instances of adaptive walks simulations in NK landscapes. The circle (respectively cross) markers corresponds to random (respectively adjacent) neighbouring epistatic interactions. Inset: Difference $\Delta f_K$ between the mean of local fitness maxima of inversions and point mutations, for random (circles) and adjacent (cross) neighbouring epistatic interactions.

where the notation $|_K$ means that the average is calculated for a fixed value of $K \in \{0, \ldots, N-1\}$.

In Fig 4 we show the behaviour of the average fitness $\langle f_v \rangle_K$, calculated from $n = 100$ instances of adaptive walks simulations in NK landscapes, with $N = 100$ and values of $K$ ranging from 0 to 99. The simulations correspond to the case of epistatic interactions with $K$ closest adjacent loci ('+' marker), and with $K$ randomly chosen loci ('○' marker). The markers of the simulations with point mutations and inversions are coloured with blue and red respectively.

For the simplest case $K = 0$, with no epistatic interaction between neighbouring loci, we can verify in Fig 4, that the average fitness for point mutations and inversions are equal $\langle f_P \rangle_{K=0} = \langle f_I \rangle_{K=0} \simeq 0.667$. In this case, the landscape is smooth with only a single peak. Hence, mutations that increase fitness are not hard to find and $\langle f_v \rangle_{K=0}$ is independent of mutation types. This result also agrees with the Kauffman's (analytical) result $\langle f \rangle_{K=0} = \frac{2}{3}$, which was calculated using order statistics arguments [48, p. 55]. Then, for $K > 0$, the average fitness $\langle f_v \rangle_K$ increases with $K$ until reaching a maximum value of fitness. In relation to this maximum, in Fig 4 we can identify the following four cases: i) point mutations with adjacent epistatic interactions, $\langle f_P \rangle_K = 0.707$ for $K = 2$; ii) point mutations with random epistatic interactions, $\langle f_P \rangle_K = 0.722$ for $K = 5$; iii) inversions with adjacent epistatic interactions, $\langle f_I \rangle_K = 0.747$ for $K = 4$; and iv) inversions with random epistatic interactions $\langle f_I \rangle_K = 0.732$ for $K = 4$. After these maximum, the fitness values decrease as $K$ increases. This trend is also consistent with the seminal simulations

carried out by Kauffman and Weinberger [46, 48]. For example, when $K \to N - 1$, the mean fitness converge to the same value, regardless of the type of epistatic interaction neighbourhood. For point mutations with random and adjacent epistatic interactions, we obtained $\langle f_P \rangle_{K=96} \simeq$ 0.580 and this agrees very well with the Kauffman's numerical outcomes for $K = 96$, c.f. [48, Tables 2.1 and 2.2]. It is worth mentioning that these trends—lower fitness being associated with increasing epistatic interactions—correspond to the well-known "complexity catastrophe" described by Kauffman [48, p. 52] (see also Refs. [52, 53]). These numerical outcomes confirm that our numerical set-up reproduces, with point mutations, what is known about $\langle f_P \rangle_K$ vs K in the NK model [46, 48]. Now, what's new is that for inversions, the average fitness values are higher than those for point mutations. Indeed, for $K \to N - 1$, the average fitness trend is very different from that of point mutations. For example, the complexity catastrophe estimates that as $K$ increases, the expected fitness of the local maximum (for point mutations) decreases toward 1/2 [48], which is indeed verified here. But for inversions, the evolutionary process reaches higher expected fitness values $\langle f_I \rangle_{K=99} = 0.610 > \langle f_P \rangle_{K=99} = 0.579$, for both random and adjacent epistatic interactions. To generalize these results, we reproduced this experiment with other, more restrictive, definition of inversion mutations. More specifically, we tested inversions on linear chromosomes (with boundary conditions) and circular inversions with upper size limit $s \leq N$, ranging from 1%—a single locus (i.e. inversions are like point mutations)—up to 100% of chromosome size. Simulations with linear chromosomes show no significant difference from our reference circular model (see supplementary S1 Text, section 2 for detailed results). Simulation with an upper size limit show that the final fitness values increases as the size limit increases. However, the gain is maximal for small $s$ values (typically up to $s = 16$) showing that small and mid-sized inversions are sufficient to reach high fitness peaks.

Fig 4 also show that, in average and for almost all $K$, the adaptive walks reach higher fitness peaks through inversions than through point mutations. To better visualise this statement, in the inset of Fig 4 we plot the following difference:

$$\Delta f_K := \langle f_I \rangle_K - \langle f_P \rangle_K,$$

for the two neighbourhoods. To specify which type of epistatic interaction we are referring to, in what follows, we will use the notation $(\Delta f_K)_{\mathrm{rnd}}$ for the case in which the fitness differences correspond to simulations with random neighbourhood, and $(\Delta f_K)_{\mathrm{adj}}$ for the adjacent ones (respectively the markers 'o' and '+' in Fig 4). In the absence of epistatic interactions, i.e. $K = 0$, we can note that $(\Delta f_K)_{\mathrm{rnd}} = (\Delta f_K)_{\mathrm{adj}} = 0$. Then, in the presence of epistatic interactions, $(\Delta f_K)_{\mathrm{rnd}}$ is monotonically increasing between $0 < K \leq 2$. Then for $2 < K \leq 31$, $(\Delta f_K)_{\mathrm{rnd}}$ is monotonically decreasing, and for $K > 31$ it is again monotonically increasing. For random epistatic interactions between $2 < K \leq 50$, the fitness values for the case with inversion are not very different from those of point mutations (note in Fig 4 that, in this interval, the red and blue curves with marker 'o' are very close to each other).

On the other hand, we can observe that for $5 < K \leq 65$, $(\Delta f_K)_{\mathrm{adj}}$ is monotonically decreasing, and for $K > 65$ it is again monotonically increasing. We can also observe that, between $K > 0$ and $K \doteq 80$, $(\Delta f_K)_{\mathrm{rnd}} < (\Delta f_K)_{\mathrm{adj}}$. Contrary to the case of random epistatic interactions, for adjacent interactions $(\Delta f_K)_{\mathrm{adj}}$ is higher since the fitness values reached by inversions are higher than those reached by point mutations (note in Fig 4 the gap between the red and blue curves with the marker '+'). So, we infer that an inversion—modifying several loci—results in a mutually advantageous conjunction with local epistatic interactions, that allows explorations of more combinations that can be beneficial. Finally, for $K > 80$, $(\Delta f_K)_{\mathrm{rnd}} \approx (\Delta f_K)_{\mathrm{adj}}$ (still monotonically increasing), i.e. regardless of the epistatic interaction neighbourhood, inversions can reach higher fitness values and attenuate the complexity catastrophe by not

decreasing towards 1/2 (compare with [48, Tables 2.1 and 2.2] and also note in Fig 4, the gap between the tails of the red and blue curves).

Therefore, our results show that in the presence of inversion it is possible to reach higher fitness when compared to adaptive walks with only point mutations.

A direct interpretation of this result is given by the properties of the inversion's mutational network as it has been described above (see e.g. Fig 2). Indeed, as it is more densely connected than the point-mutation mutational network, it is likely to allow a larger exploration of the fitness landscape and thus reach higher peaks, as observed here. However, given that the ruggedness of a fitness landscape depends on the mutational operator at work, an alternative explanation is that inversion mutations result in a smoother fitness landscape than that of point mutations, hence facilitating the finding of trajectories leading to higher peaks.

To test this assumption, we generalized the roughness measure introduced by Aita, Ikamura and Husimi in [54]. More precisely, we measured deviations from fitness additivity (in the language of the NK model, we say that a landscape is additive when it is non-epistatic, that is, $K = 0$). We here use the term roughness from [54] to distinguish this measure, which is a local one, from the classical ruggedness of the NK-fitness landscape which is a global property of the landscape. See, for example, Refs. [55] and [56], for other definitions of roughness and how they are calculated. Following the approach introduced in [54], we computed the roughness of the fitness landscape as the root mean square fitness variation due to each possible mutation, for both point mutations and inversions (see Methods for a formal mathematical definition of this measure). The results are shown in Fig 5. As expected, for point mutations

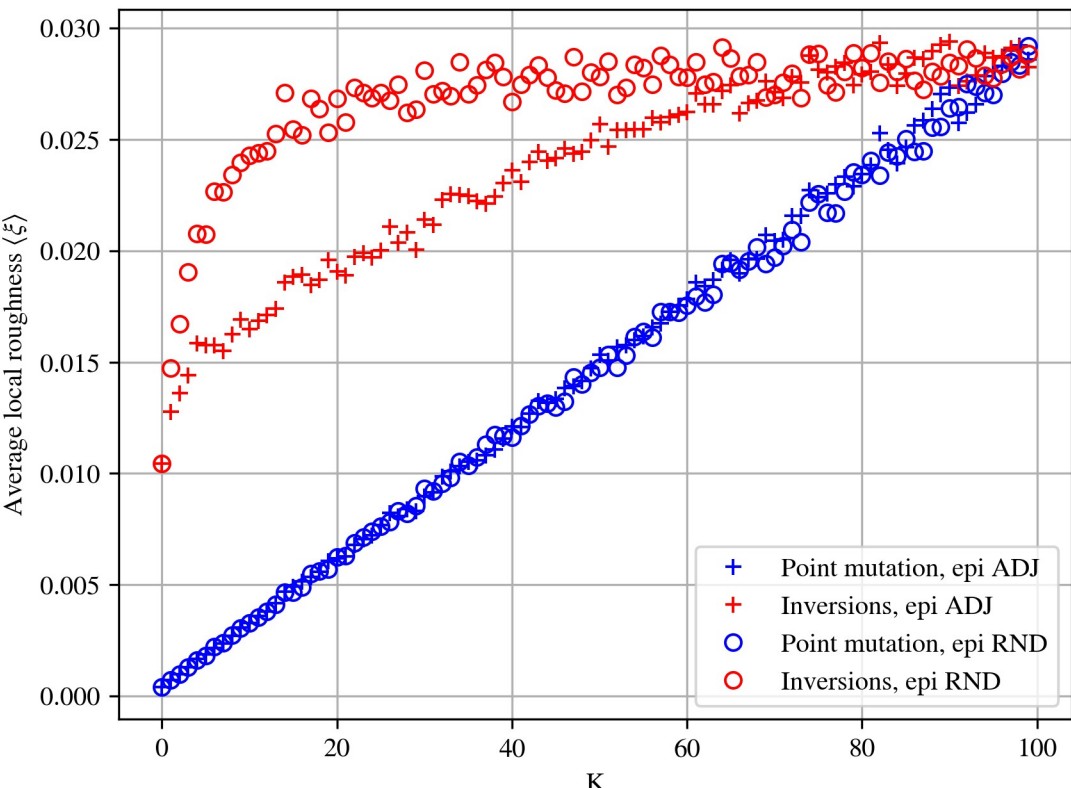

**Fig 5. Average value of the local measure of roughness.** Local roughness measured as the mean square differences between the fitness in a point of the landscape and its neighbours, for inversion (red) and point mutations (blue), for adjacent (crosses) and random (circles) epistatic interaction neighbourhoods, averaged for 100 instances of NK fitness landscapes.

the roughness of the fitness landscape is (almost) linearly proportional to the epistatic interaction parameter $K$, for both types of epistatic interaction neighborhoods (adjacent and random). In contrast, in the case of inversion mutations, the roughness is always greater than that of point mutations, this trend being particularly visible for random epistatic neighborhood when compared to adjacent neighborhood. Interestingly, even for $K = 0$, the roughness is already higher than for point mutations (both epistatic neighborhood being equivalent in that case) while for $K = N - 1$ the roughness converges approximately towards similar values both for inversions and point mutations whatever the epistatic neighborhood.

This result shows that inversion mutations actually don't smooth the fitness landscape. On the opposite, the average roughness increases much faster with $K$ in the case of inversions than in the case of point mutations (the roughness of the inversion-based fitness landscape with $K = 1$ being similar to the one for point mutations with $K = 50$, Fig 5). This result also suggests a new explanation for the advantage of inversion mutations over point mutations. While the high connectivity of the inversions mutational network enables a better exploration of the fitness landscape, this effect is hampered (and not facilitated) by the effect of inversions on the roughness and while the combination of both positive (connectivity) and negative (roughness) is favorable for all values of $K$ in the case of adjacent neighborhood, it is only favorable for high values of $K$ in the random epistatic neighborhood. Indeed, for epistatic interactions with a random neighborhood, there is no noticeable difference between the average fitness values up to $K \simeq 40$ (red and blue curves with markers '○' in Fig 4). This is likely to be due to the fact that inversions are segmental operators. When epistatic interactions are confined to a segment close to the focal nucleotide (which is the case for the adjacent neighborhood but not the random one), both segments can largely overlap, hence limiting the effect of the inversion to a set of epistatically interacting genes. This reduces the average roughness (compared to random epistatic neighborhood), leading to a more efficient exploration of the fitness landscape. Although a full mathematical proof is out of the scope of this paper, we develop a representative mathematical analysis that illustrates the origin of this pattern in S1 Text (section 3).

## Discussion

The results presented in this paper show that intragenic inversion mutations lead adaptive walks to reach higher fitness peaks on rugged landscapes. We have performed simulations in NK landscapes and have shown that the expected fitness values are higher for inversions than for point mutations. This holds for all degrees of ruggedness (epistasis), ranging from single-peak (K = 0), moderately rugged ($1 \leq K < N - 1$), to fully rugged landscapes (K = N − 1). Simulations with point mutations agreed well with the already known characteristics of the NK model, and made it possible to establish a "control group" to ensure the reliability of the differences with inversion mutations. We also observed that for adjacent epistatic interactions, the differences of expected fitness values between inversions and point mutations are greater than in the case of random epistatic interactions. In this sense, we conjecture that this should be the consequence of a synergistic effect between inversions and adjacent epistatic interactions, epistatic adjacency enabling a set of interacting loci to be inverted at once without affecting other, non-interacting, loci (see S1 Text for a detailed discussion). We believe that the relationship between epistasis and structural inversions is an area that has not yet been deeply explored.

Our analysis consisted of adaptive processes driven by mutation-specific evolutionary steps, i.e. in addition to the widely used point mutations, we introduced a minimal model of inversion mutations in double-stranded (digital) genomes. This model has also revealed some consequences of the limits of simulations with point mutations. We showed that, in addition to ruggedness due to epistatic interactions, the escape process can also depend on the

interrelationships between the genotype space and the fitness landscape that are mediated by the type of mutation. In particular, we showed that for inversion mutations, the graph-theoretical representation of accessible mutants displayed a complex topology, in comparison with the canonical genotype space constructed with point mutations. By definition, the node degree of the graph of mutations is the number of accessible mutants. In the case of inversions this number is no longer constant over the node set—as in the case of point mutations—but varies depending on the specific sequence composition. Therefore, although it is correct that we can generate genotypic space through point mutations, that does not (strictly) imply that evolutionary paths in the fitness landscape have to be solely through point mutations. In this sense, the inversion mutations allowed us to reveal new topological properties of the interconnection between genotypes through what we have defined as mutational networks. Indeed, it is this mutational network that mediates the interconnection between genotype space and the adaptive dynamics in the fitness landscape. The mutational network can be translated as a fitness network when the fitness values of each mutant genotype are included. Thus, revealing the directions of possible evolutionary paths. Let us remark that graph theory is a framework used in various models of evolution (see for example Refs. [57–67]) and has been advantageous in analysis that use the Kauffman model, such as [51, 68–72]. However, most of these graph representations for mutations and fitness landscapes are isomorphic to the (hypercubic) genotype space, whereas our definition of mutational and fitness networks are not necessarily isomorphic to the genotype space. Moreover, we showed that for fitness networks generated by inversions, there are more mutational pathways between genotypes compared to point mutations. Therefore, for inversion mutations, at each step an evolutionary process can potentially explore more accessible mutants, than the "classical" estimation $(|\mathcal{A}_\mathcal{L}| - 1)N$, for any alphabet $\mathcal{A}_\mathcal{L}, \forall \mathcal{L} \in \{2n : n \in \mathbb{N}\}$, with size $|A_\mathcal{L}| = 2n$. In this sense, our work can straightforwardly complement the results reported in [73], for $|\mathcal{A}_\mathcal{L}| > 2$ with point mutations. The main message here is that in addition to the well-known utility of the fitness landscape metaphor, its topographic properties (due to epistasis) are not sufficient for modelling the escape from local peaks, and additional information should be included via the topology of mutational networks and fitness networks. This information can be useful to predict evolutionary trajectories in fitness landscapes [44, 55, 56, 74–76].

An important takeaway from this work is that an effort must be made to incorporate features of the structural variations of genomes, such as submicroscopic intragenic rearrangements. In this paper, we have taken a first step in this direction by modelling inversion mutations. A by-product of the construction of our model of inversion mutations revealed topological properties associated with the genotypic space and the accessible mutants for potential evolutionary paths. This is a consequence of the combinatorics of accessible mutants, which depends on structural aspects of this type of chromosomal rearrangement. On the other hand, our graph theoretical representations are consistent with the idea of adaptive walks in complex networks [16, 17, 77]. The key difference is that in our model we do not have to postulate *a priori* a network that satisfies a certain topology (e.g. scale-free or random) as in [16, 17, 77]. In our case, the topology arises as a consequence of the type of mutations. Following the interpretation provided in [17] about multiple mutations in a single evolutionary step, we suggest that another alternative to justify (or interpret) their *topological inspired walks* is via generic structural variations in concomitance with our model. In the case of the simulations of adaptive walks on complex networks reported in [77], the authors state that "it seems more realistic to ponder sequence spaces where the node's connectivity is not the same for every node, as it is in hypercubes". We agree with the authors of [77] and [17] that the degree of a genotype (in the mutational network for us) measures the availability of accessible mutants.

This is what we have proposed to define as mutability, that is, the ability to change from one genotype to another under a mutation. We believe that it could be interesting to explore this notion of mutability and its relationship with the genetic potential of mutations that give rise to novel (beneficial) phenotypes [78].

In this work, we studied the effect of inversion mutations on the maximum fitness reached on rugged landscapes and compare it to the maximum fitness reached by point mutations. Importantly, both kind of mutations have been tested in independent simulations under the Strong Selection Weak Mutation regime. Hence, although we have shown that inversions reach higher fitness values in these conditions, we cannot compare the evolutionary dynamics between these two mutational setups. However, it is important to stress that, in a real population, both kind of mutations would not occur in isolation. Any evolving population undergoes all mutation types, including both point mutations and inversions. Hence, inversion mutations should not be considered in competition with point mutations, but rather as a synergistic interaction. Studying the consequences of this interaction on the evolutionary dynamics constitutes one of the most exciting perspective of this work.

In our model we did not include recombination [79] and other chromosomal rearrangements, such as duplications, deletions, and translocations. However, our purpose with inversion mutations has been to exemplify a simple mechanistic model of structural variation. Of course, our sequence model includes many simplifications. In particular, we use binary sequences and simulated a fully coding compact genome with a circular double-stranded DNA. Although most of our theoretical results hold in a more general case, the effect of inversions on more realistic coding sequences (with e.g. a 4 bases alphabet, multiple reading frames and ORF identified by start and stop codons and separated by non-coding sequences) could reveal other properties of interest. For instance, micro-inversions effect on reading frames is likely to be specific and very different from the effect of point mutations (that don't shift the reading frames) or InDels (that are likely to shift it). Indeed, inversions can alter a subsequence of an ORF without changing the reading frame of the start/stop codons. On the opposite, an inversion can easily remove (or create) a stop codon.

All throughout this study, we focused on binary sequences, simplifying the 4-bases nature of real genomes. Although the binary description is very common in theoretical and computational models, it is important to mention that the properties of the different mutational operators, hence of the generated mutational networks, may differ depending on the size of the alphabet [73]. In the case of inversions, although our main conclusions about the complex structure of the mutational network still hold, a 4-bases alphabet with two pairs of complementary bases (`A`-`T` and `C`-`G`) would introduce important properties compared to the binary case. Indeed, given the mathematical definition of inversions (Eq 5), it is straightforward that the composition of the inverted segment will conserve the relative fraction of `AT` and `CG` pairs relatively to the original one (a specific situation being the inversion of a segment of size one that can only switch `A` and `T` or `C` and `G`). It immediately follows that inversions cannot change the `AT`/`CG` ratio of the sequence and that, for a sequence of length $N$, the mutational network generated by inversions contains at least $N + 1$ disconnected sub-networks (which sizes will depend on the `AT`/`GC` ratio of the sequence, strongly biased sequences leading to smaller sub-networks). Hence, compared to the binary model, the 4-bases model increases the size and connectivity of the mutational network generated by point mutations [73]. On the opposite, in the inversion-generated mutational network, it isolates several sub-networks from each others. The effect on evolutionary dynamics, as we studied it here using the NK landscape, is still to be explored. Indeed, depending on the composition of the initial sequence, with a 4-bases alphabet some local/global optima may not be accessible. Consequently, the advantage of inversion mutations may be reduced, and even be cancelled if the number of local optimum is very low

(i.e. for $K \ll N$). However, it is worth mentioning that real genomes undergo both kinds of mutational events and that point mutations connect the inversion-isolated sub-networks by changing the AT/GC ratio of the sequence. Exploring how both kinds of mutations (and others) interact is clearly beyond the scope of this manuscript, but studying the synergistic effect of inversions and point mutations in a more sequence-realistic model like the Aevol model [80, 81] is clearly an appealing perspective.

Despite the simplifications used in this study, our results show that structural inversions could be considered not only as changes in the orientation of sequences that don't alter the genetic content, as classically supposed in the literature [39], but also as a source of intragenic variations. In this sense, our phenomenological model is supported by the empirical evidence of an intragenic inversion associated with the creation of new regulatory elements—required e.g. for the termination-activation of transcription in the nitric oxide synthase gene in *Lymnaea stagnalis* [30]. As well, the pathogenic mutation due to an intragenic inversion of seven nucleotides in human mitochondrial DNA [29]. Furthermore, although first generation sequencing technologies were unable to identify submicroscopic rearrangements [82], the development and availability of novel sequencing technologies [28, 82], opens the possibility of characterising intragenic structural variations and may be of particular importance to unravel new aspects of mutations in molecular evolution. Therefore, we may soon require new theoretical and computational models to simulate the fullness of chromosomal rearrangements in evolutionary biology. We hope this work makes a first step in this direction.

## Conclusion

The statements presented in this paper provided computational evidence that, for a very simple model of evolution in the strong selection weak mutation limit, an adaptive process in rugged landscapes driven by intragenic inversion mutations can reach higher fitness values (compared to a same process driven by point mutations). Therefore, this implies that intragenic inversion mutations can lead evolution to escape local fitness peaks in rugged landscapes. The way our model was conceived also proves that escape from a local peak of fitness can occur in constant environments without contingencies. Our model for inversion mutations not only elucidated an escape mechanism, but have also made it possible to uncover interesting aspects about the combinatorics of inversions and their relationship with mutated genotypes, genotype spaces and fitness landscapes in terms of graphs representations.

## Methods

### The model

**Preamble: Limits in the single-nucleotide mutation scenario.** It is worthwhile to state the main issue when point mutations are the only source of genetic variations in evolutionary models. At the molecular level and besides the fitness values, the structure of DNA mutations constrains the way evolution can move through the genotype space. For example, a common reasoning in molecular evolution theory is: for any alphabet $\mathcal{A}_{\mathcal{L}}, \forall \mathcal{L} \in \{2n : n \in \mathbb{N}\}$, with size $|A_{\mathcal{L}}| = 2n$ (this total number of letters with even parity being due to the double-stranded structure), a sequence $\mathbf{x}$ of $N$ nucleotides, would have

$$D(\mathbf{x}) = (|\mathcal{A}_{\mathcal{L}}| - 1)N \tag{1}$$

mutant neighbours differentiated by a single point mutation [45, 49, 83, 84]. These $D$ neighbouring genotypes are available for natural selection. Then, at the SSWM limit, only one of these neighbouring sequences can be fixed, chosen among those with fitness values higher

than the wild-type fitness. Once a new mutant is fixed, a new set of $D$ mutant neighbours is available for selection. Repeating this process unfolds an evolutionary path until it reaches a local (or global) fitness peak in the rugged landscape. However, if a local peak is reached (i.e. the state when all $D$ accessible genotypes have strictly lower fitness values), then the evolutionary process is "trapped" because any other fitter genotype is at two or more point-mutational steps away (i.e. only attainable by "descending" through a valley in the fitness landscape).

**Digital sequence scheme.**   First, let us recall the very basic and well-known notion that DNA is a double strand molecule with two nucleotides chains, held together by complementary pairing of adenine (A) with thymine (T) and guanine (G) with cytosine (C). Given a DNA strand, as for example ATCGATTGAGCTCTAGCG, its complementary strand is TAGCTAACTC GAGATCGC, which in the IUPAC's notation is

$$5' - \mathtt{ATCGATTGAGCTCTAGCG} - 3'$$
$$3' - \mathtt{ƆƆⱯⱯƆ⊥ⱯⱯƆ⊥ƆƆⱯⱯ⊥Ɔ⊥Ɔ} - 5'\ '$$

where the leading strand is on top and the DNA strand orientation is by convention $5' \rightarrow 3'$.

Throughout the presentation of our model, we adopt the alphabet $\mathcal{A}_2 = \{0, 1\}$, so the genotypes are binary sequences of (constant) length $N \in \mathbb{N}_{\geq 2}$. As a low-level structural representation consistent with the DNA molecular biology, these genotypes are double-stranded sequences

$$\mathbf{x} \coloneqq x_1 x_2 \cdots x_{N-1} x_N,$$

with $N$ digital nucleotides $x_i \in \mathcal{A}_2, \forall i \in \{1, \ldots, N\}$, where the complementary sequence $\bar{\mathbf{x}}$ is defined such that $\forall x_i \in \mathcal{A}_2, \bar{x}_i \coloneqq 1 - x_i, \forall i \in \{1, \ldots, N\}$. All the sequences $\mathbf{x}$ of size $N$ define the set $\mathcal{X} \in \{0, 1\}^N$ of possible genotypes.

In analogy with the example above, the representation of this double-stranded digital sequence is:

$$\mathtt{011001100011110010}$$
$$\mathtt{100110011100001101}\ .$$

It is very important to clarify that our schematic representation should not be confused with the usual encoding $\mathcal{A}_{\mathrm{DNA}} \rightarrow \mathcal{A}_2$, with the convention purines $\{\mathtt{A}, \mathtt{G}\} \rightarrow 0$ and pyrimidines $\{\mathtt{T}, \mathtt{C}\} \rightarrow 1$. The artificial genetics in our model only have two nucleotides, and they complement each other. We also adopt the following approximations:

- We assume that the sequences are circular, i.e. periodic boundaries: $x_{N+i} = x_i, \forall i \in \{1, \ldots, N\}$.

- We neglect the existence of coding sequences separated by non coding regions. Biologically, this corresponds to compact genomes with almost no non-coding sequences. In this sense, our model mimics the molecular evolution of some viruses [40] or (animal) mitochondrial DNA [43]. Consequently, we are modelling intragenic mutations [44, 82].

- We neglect geometrical aspects such as physical configurations like folding, twists, coiled structures, hairpin loops, etc.

- We do not consider transcription-translation processes.

- We do not include recombination, so we are modelling asexual replication.

**Structural inversions model.** To establish the main idea of DNA inversion mutations well, now let us illustrate this mechanism with the following representation:

$$
\begin{aligned}
&5' - \text{ATC}\,\boxed{G}\,\text{ATTGAGCTC}\,\boxed{T}\,\text{AGCG} - 3' \\
&3' - \text{CGCT}\,\boxed{C}\,\text{TAACTCGAG}\,\boxed{A}\,\text{TCGC} - 5'
\end{aligned}
$$

$$
\begin{aligned}
\cdots\,\boxed{G}\,\text{ATTGAGCTC}\,\boxed{T}\,\cdots & \xrightarrow[\circlearrowleft]{inversion} \cdots\,\boxed{A}\,\text{GAGCTCAAT}\,\boxed{C}\,\cdots \\
\cdots\,\boxed{C}\,\text{TAACTCGAG}\,\boxed{A}\,\cdots & \qquad\qquad \cdots\,\boxed{T}\,\text{CTCGAGTTA}\,\boxed{G}\,\cdots
\end{aligned}
\tag{2}
$$

$$
\begin{aligned}
&5' - \text{ATC}\,\boxed{A}\,\text{GAGCTCAAT}\,\boxed{C}\,\text{AGCG} - 3' \\
&3' - \text{CGCT}\,\boxed{T}\,\text{CTCGAGTTA}\,\boxed{G}\,\text{TCGC} - 5'
\end{aligned}
$$

where in the middle is depicted the segment where the mutation occurs and their corresponding inversion (boxes and colours highlight the segment where the inversion occurs). A glance over schema (2) shows why a double-stranded-like model is unavoidable to model intragenic inversions.

For computational purposes, an inversion mutation can be split in two operations:

- The conjugation operation $\hat{C}$:

$$
\hat{C} : x_i, x_{i+1}, \ldots, x_{j-1}, x_j \longrightarrow \bar{x}_i, \bar{x}_{i+1}, \ldots, \bar{x}_{j-1}, \bar{x}_j,
\tag{3}
$$

for $i, j \in \{1, \ldots, N\}$.

- The permutation operation $\hat{P}$:

$$
\hat{P} : x_i, x_{i+1}, \ldots, x_{j-1}, x_j \longrightarrow x_j, x_{j-1}, \ldots, x_{i+1}, x_i,
\tag{4}
$$

for $i, j \in \{1, \ldots, N\}$.

Note that, as the genome is circular, there is no relation of order between $i$ and $j$ (i.e. $\hat{C}$ and $\hat{P}$ are well-defined even when $i > j$). Other sites $k \notin \{i, \ldots, j\}$, remain unchanged. Then, we have the (two-step) inversion operation:

$$
\hat{I} \equiv \hat{C} \circ \hat{P}.
\tag{5}
$$

For example:

$$
\begin{aligned}
&011\,\boxed{0}\,011000111\,\boxed{1}\,0010 \\
&100\,\boxed{1}\,100111000\,\boxed{0}\,1101
\end{aligned}
\xrightarrow{\hat{I}=\hat{C}\circ\hat{P}}
\begin{aligned}
&011\,\boxed{0}\,000111001\,\boxed{1}\,0010 \\
&100\,\boxed{1}\,111000110\,\boxed{0}\,1101
\end{aligned}.
\tag{6}
$$

Trivially, it can also be verified that $\hat{C}$ and $\hat{P}$ commutes:

$$
\hat{C} \circ \hat{P} = \hat{P} \circ \hat{C}.
\tag{7}
$$

Besides, Eq 5 can also define a single-locus mutation: when $i = j$, then $\hat{I}$ is a single bit-flip i.e. a point mutations.

Computationally, these operations can be easily implemented through Algorithm 1: `Mutate`. With this simple algorithm, it is possible to calculate the combinatorics of the inversion mutations. For example, the enumeration of accessible mutants for each genotype with $N = 4$ shown in Fig 1.

**Algorithm 1:** `Mutate` $(\mathbf{x}, i, j, N)$

```
input: x ∈ 𝒳, [i, j] ∈ {1, ..., N}
  1: l ← i
  2: u ← j
```

```
3: y ← x
4: repeat
  5: y₁ ← (1 − xᵤ)
  6: l ← l + 1 mod N
  7: u ← u − 1 mod N
8: untill l = j + 1
9: return y ∈ {0, 1}ᴺ
```

## NK model

A well known model of genetic epistatic interactions is the NK family of rugged multipeaked fitness landscapes [45–48]. In this model, besides the genome length $N \in \mathbb{N}$, the integer $K \in \mathbb{Z}_{(0, N-1)}$, describes the epistatic interactions between loci in the genome and the contribution of each component to the total fitness, which depends on its own value as well as the values of $K$ other loci. The fitness per locus is formally defined as:

$$f_i : \{0, 1\}^{K+1} \longrightarrow [0, 1), \quad \forall 1 \le i \le N.$$

Here $f_i(x_i, x_{i_1}, \dots, x_{i_K})$ depends on the state of locus $x_i \in \{0, 1\}$ and $K$ other loci $x_{i_K} \in \{0, 1\}$. The $f_i$'s are given by $N \cdot 2^{K+1}$ independent and identically distributed random variables sampled from a given uniform probability distribution. See the example shown in [47, Table I], for a very illustrative description for the computing of the epistatic contribution per locus. The pattern into which the scheme of interaction between loci is connected is known as the epistatic neighbourhood [46, 47]. In our simulations we use two popular neighbourhood models:

- The adjacent neighbourhood model, where $i$ and the $K$ other sites are successively ordered, i.e. $i, i+1, \dots, i+K$ (each variable modulo $N$ when using periodic boundary conditions).

- The random neighbourhood model, where $i$ and the $K$ other loci are chosen at random according to a uniform distribution from $\{1, 2, \dots, N\}$.

Examples for $N = 4$ are depicted in Fig 6.
The total fitness $f \in [0, 1)$ for the genotype $\mathbf{x} \in \mathcal{X}$ is then defined as:

$$f(\mathbf{x}) := \frac{1}{N} \sum_{i=1}^{N} f_i(x_i, x_{i_1}, \dots, x_{i_K}), \tag{8}$$

where $\{i_1, \dots, i_K\} \subset \{1, \dots, i-1, i+1, \dots, N\}$.

The most important feature of the NK model is that the parameter $K$ tunes the landscape ruggedness, that is the distribution of fitness local maximums, ranging from non-epistatic interactions when $K = 0$ (a Mount-Fuji-like landscape with a single peak), to the full rugged (or random) landscape when $K = N - 1$.

## Adaptive walks

The zero-order approximation of our model to population genetics theory is on the limit of strong selection weak mutation (SSWM) [49, Ch. 5]. In this limit, the adaptive walk model describes very well the molecular evolution of isogenic (monomorphic) populations, as the sequential fixing of novel beneficial mutations. Therefore, the simulation of evolutionary processes in our digital model can easily be translated within this framework. That is, instead of describing a population of organisms with a pool of genotypes, it is sufficient to simulate the evolutionary trajectory over the fitness landscape of a single initial genotype and its successive mutations.

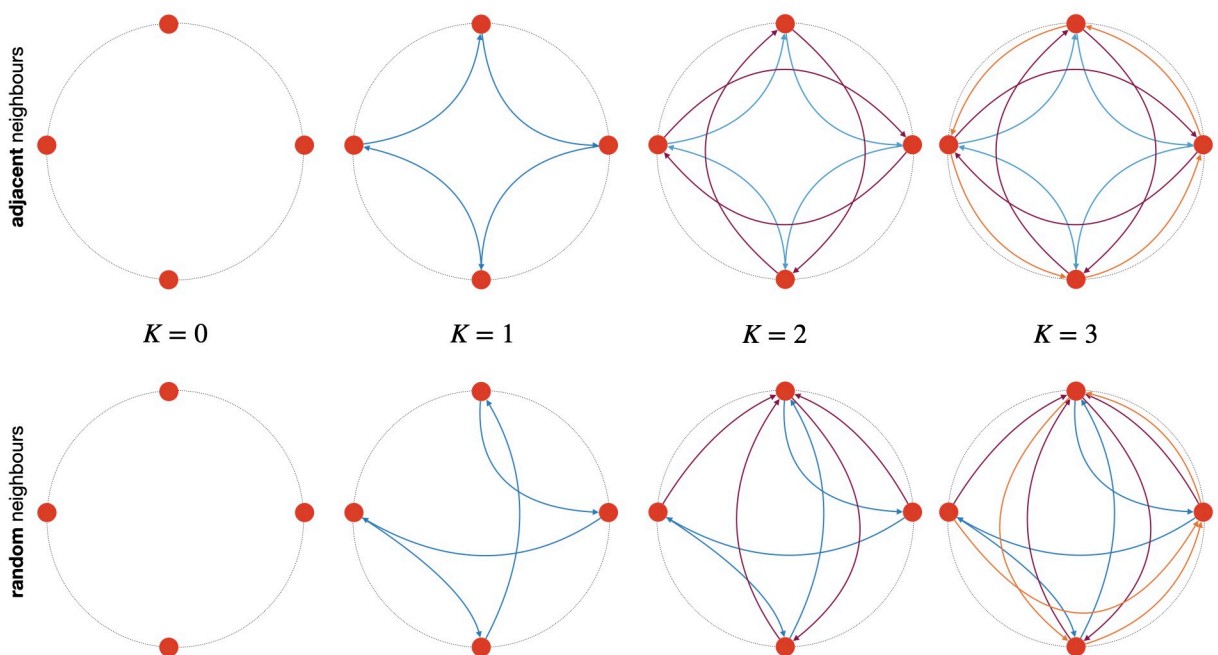

**Fig 6. Neighbourhood epistatic interactions of the NK model.** Schematic representation of a circular genome with $N = 4$ and epistatic interactions $K = \{0, 1, 2, 3\}$. Top: Example of adjacent neighbours. Bottom: Example of random neighbours (let us remark that in this case, for $K > 0$ the in and out degrees per node do not have to be equal).

The procedure goes as follows: a (randomly chosen) starting genotype $\mathbf{x} \in \mathcal{X}$ varies through successive mutations (calculated with Algorithm 1: Mutate) resulting in a mutated sequence $\mathbf{y} \in \mathcal{N}_v(\mathbf{x})$, where $\mathcal{N}_v(\mathbf{x})$ is the set of all accessible mutants of genotype $\mathbf{x}$. Then, the fitness $f(\mathbf{x})$ and $f(\mathbf{y})$ are calculated according to the NK model with Eq 8. If $f(\mathbf{y}) > f(\mathbf{x})$, the mutated genotype is selected, otherwise other mutations on $\mathbf{x}$ are tested until the fitness increases. In Algorithm 1: Mutate, the loci $[i, j] \in \{1, \ldots, N\}$ are drawn from a pseudo random number generator function. With this recipe, the evolutionary dynamics is simulated up to a local fitness maximum is reached, i.e. $\mathbf{x} \in \mathcal{X}$ satisfies: $f(\mathbf{y}) < f(\mathbf{x})$, $\forall \mathbf{y} \in \mathcal{N}_v(\mathbf{x})$. In other words, we verify that all mutants for a given genotype do not have higher fitness values, if not, the simulation continues.

The main routine to simulate adaptive walks on the Kauffman's NK-fitness landscape model, with point mutations (as usual) and inversions (as new) is available at https://gitlab.inria.fr/letrujil/getting-higher. Our code is based on the one developed by Wim Hordijk (in its version of August 23, 2010 and which is available at http://www.cs.unibo.it/∼fioretti/CODE/NK/), which uses some code from Terry Jones (https://github.com/terrycojones/nk-landscapes).

## Roughness measure

The roughness to slope ratio proposed by Aita, Iwakura, and Husimi in [54], can be re-interpreted in terms of the local measure of roughness of the surface of a solid material or an irregular interface, i.e. as the root mean square surface width in function of the height at a given place on the surface (see for example [85, p.22]). In our case if we assume that the "height" is equivalent to the value of the fitness $f(\mathbf{x})$ of genotype $\mathbf{x}$, and "the place on the surface" corresponds to a domain (on the surface) of the fitness landscape, then we can define the measure

of local roughness given a genome $\mathbf{x} \in \{0, 1\}^N$ as:

$$\xi_v(\mathbf{x}) := \left( \frac{1}{|E(m_v(\mathbf{x}))|} \sum_{(\mathbf{x},\mathbf{y}) \in E(m_v(\mathbf{x}))} (f_v(\mathbf{x}) - f_v(\mathbf{y}))^2 \right)^{\frac{1}{2}}, \forall \mathbf{x} \in \{0, 1\}^N, \tag{9}$$

where the index $v$ denotes point mutations (P) or inversions (I), and $E(m_v(\mathbf{x}))$ is the set of all possible mutations $\mathbf{y}$ of a given genotype $\mathbf{x}$ (i.e. the edges of the directed multigraph of mutations $m_v(\mathbf{x})$).

## Supporting information

**S1 Fig. NK fitness networks for epistatic interactions with adjacent neighbouring.** Representative instances of the NK model for $N = 4$ and their fitness networks in layered representation. The layers are constructed such that each node is assigned to the first possible layer, with the constraint that all its predecessors must be in earlier layers. The colors of the nodes correspond to the values of the out-degrees, i.e. the number of edges going out of a node (note that color scales differ in range between panels). Therefore, nodes with node out-degrees equal to zero correspond to local fitness maxima (sink nodes). The landscapes' ruggedness are: single peaks $K = 0$, intermediate ruggedness $K = 1$ and full rugged case $K = 3$, for adjacent neighbouring epistatic interactions. Node sizes are scaled with fitness values (best fitness, largest and vice versa). Global maximum of fitness are encircled in red. While the global minimum in blue. The total number of fitness maxima and minima are also reported. See 3 in main text for epistatic interactions with random neighbouring.
(TIFF)

**S1 Text. Fraction of invariant inversions, linear inversions and synergistic effect of inversions.**
(PDF)

## Acknowledgments

We wish to acknowledge insightful conversations with members of the Beagle team, especially David P. Parsons and Aoife O. Igoe also for their suggestions on the manuscript. L.T. thanks the Institut National des Sciences Appliquées (INSA) as well as the Laboratoire d'InfoRmatique en Image et Systèmes d'information (LIRIS) for hospitality while part of this research was done and would like to thank Anton Crombach, Harold P. de Vladar and Ivan Junier for useful discussions and valuable support. P.B. is grateful to Laurent Turpin and Nathan Quiblier for stimulating discussions.

## Author Contributions

**Conceptualization:** Leonardo Trujillo, Guillaume Beslon.

**Data curation:** Leonardo Trujillo, Paul Banse.

**Formal analysis:** Leonardo Trujillo, Paul Banse, Guillaume Beslon.

**Investigation:** Leonardo Trujillo, Paul Banse, Guillaume Beslon.

**Methodology:** Leonardo Trujillo, Paul Banse, Guillaume Beslon.

**Project administration:** Leonardo Trujillo.

**Resources:** Guillaume Beslon.

**Software:** Leonardo Trujillo, Paul Banse, Guillaume Beslon.

**Supervision:** Leonardo Trujillo, Guillaume Beslon.

**Validation:** Leonardo Trujillo, Paul Banse, Guillaume Beslon.

**Visualization:** Leonardo Trujillo, Paul Banse.

**Writing – original draft:** Leonardo Trujillo.

**Writing – review & editing:** Leonardo Trujillo, Paul Banse, Guillaume Beslon.

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
