## [Decision Letter · Decision Letter 0]

8 May 2022

Dear Dr. TRUJILLO,

Thank you very much for submitting your manuscript "Getting higher on rugged landscapes" for consideration at PLOS Computational Biology.

As with all papers reviewed by the journal, your manuscript was reviewed by members of the editorial board and by several independent reviewers. In light of the reviews (below this email), we would like to invite the resubmission of a significantly-revised version that takes into account the reviewers' comments.

The manuscript has been evaluated by three expert reviewers. They all agree that the manuscript makes an interesting contribution. At the same time, all reviewers make a number of (extremely constructive) suggestions about how to further improve the quality of the paper. Please take these suggestions into account.

We cannot make any decision about publication until we have seen the revised manuscript and your response to the reviewers' comments. Your revised manuscript is also likely to be sent to reviewers for further evaluation.

Sincerely,

Christian Hilbe

Associate Editor

PLOS Computational Biology

James O'Dwyer

Deputy Editor

PLOS Computational Biology

The manuscript has been evaluated by three expert reviewers. They all agree that the manuscript makes an interesting contribution (based on my own reading, I concur with them). At the same time, all reviewers make a number of (extremely constructive) suggestions about how to further improve the quality of the paper. I would like the authors to take these suggestions into account.

Let me also note that I agree with reviewer #3 regarding the title. As is, the title seems a bit too generic. I would like to encourage the authors to find a title that more explicitly states that the mechanism considered here is based on inversions.

Reviewer's Responses to Questions

**Comments to the Authors:**

Reviewer #1: In this manuscript, the authors investigated how populations can explore the genotypic space using the fitness landscape framework. The usual approach to this question is to use point mutations. However, the authors chose here to use a different type of mutations: small structural variants and inversions in particular. Including these larger mutations change the connectivity of the genotype space, allowing for “long range” jumps across the genotype space and therefore the fitness landscape. In particular, it means that such variants are more effective at exploring the fitness landscape and less likely to end up on a local fitness pic. The authors therefore present here an alternative explanation to the classical problem of “crossing fitness valleys”.

I enjoyed reading this manuscript. The idea explored here is really interesting.

One question I have that is not answered is how the choice of modelling a small circular chromosome (at most 100 bp) affects the results obtained here. Using a linear chromosome should reduce the connectivity between genotypes, but it is unclear how strong the effect would be.

I could not find any information on the distribution of inversion length used for the simulations. To my understanding the simulations were conducted allowing the inversions to span from 1 bp to N bp. I wonder what happens if the inversion size were restricted to a certain size range (e.g. 10 to 30% of the chromosome).

The authors are using the weak mutation strong selection approximation. This means that a mutation that has an infinitely small chance of appearing, will still appear with certainty if it is the only beneficial mutation left. This assumption means that the exploration of the fitness landscape under this regime and a more ”realistic” exploration of the fitness landscape may yield different results. In particular, since the connectivity of the fitness landscape increases (a lot) in the presence of inversions, it means that the ratio beneficial/deleterious mutations is much smaller for genotypes close to the optimum (than in the point mutation scenario – as it has access to many more “bad” options). I therefore wonder how the use of inversion may affect the waiting time for beneficial inversions, compared to point mutations. Therefore I believe it will be interesting to characterize the distribution of fitness effects of both SNPs and inversions, and the waiting time to the next “step”.

It would also be interesting to compare the dynamics of the two populations (one using point mutations, the other inversions), and see which one increases first in fitness, using parameters similar to WMSS regime – but allowing populations to naturally evolved. For competition, it is better to be fitter now than to be the fittest latter. Therefore, while the process described here is interesting, it is hard to judge its evolutionary relevance without this information.

Finally, one biological aspect that is ignored here is that the new mutations, generated by the inversion, are not random and are conditioned by the ancestral state. For example, the three stop codons will stop being stop codons if inverted, and a leucine, phenylalanine and a serine will become stop codons to point out the most obvious example. I believe exploring this potential aspect is beyond the scope of the manuscript but it would be interesting if the authors could comment on this, as the NK model may fail to capture this effect.

I believe Figure 3 would be better served by showing less cases (moving them to a supplementary figure) but with remaining figures better annotated. In addition, according to the legend “Node colours correspond to fitness values, increasing from left to right.”, so both colors and positions along the x axis indicate fitness? In that case, I do not understand why points of different colors are on the same vertical line.

Minor comments:

Line 92: this is a bit misleading here. The only example given above of a submicroscopic inversion was the 7bp inversion in the mitochondria. Reference 54 here refers to inversions that are between 1 and 3kb. It is roughly 3 orders of magnitude of difference.

Line 134-135. Should N not be excluded in the inequality? If the inversion spans the whole chromosome, then 5’-3’ becomes 3’—5’ and vice versa and nothing has changed.

Line 200-201. I do not understand this sentence.

Reviewer #2: Trujillo et al. present an interesting generalization of the study of adaptive walks

in which the main mechanism of producing variation is through inversion mutations instead of

point mutations as usual.

As we know the adaptive walk framework holds in the strong selection weak mutation regime. In this limit, adaptation is modelled

as the exploration of the genotypic space through point mutations, and once a local optimum of the fitness landscape is reached

the process is halt. Conceptually, the problem of the adaptive walk modifies as the mechanism of variation changes. The definition of local optimum

is made with respect to the mechanism of genetic variation.

Through extensive computer simulations, the authors show that higher fitness values are reached through inversion mutations. The effect is even more prominent

when epistasis occurs among adjacent locus of the genome.

I think the authors bring an important contribution to this topic. Certainly, this contribution will

shed new light into this problem and motivate future contributions of the concerning other mechanisms of genetic variation.

I think the contribution can be accepted for publication after some important points are raised in a revised version of the manuscript.

Points to be addressed:

In order to understand the effectiveness of the mechanism of inverse mutations when compared to point mutations, we need a fair comparison of the problems

that are conceptually distinct, as highlighted above. Important measurements to be considered and compared:

1) What is the density of local optima under the mechanism of inversion mutations for the same set of parameters used for point mutations (same N, same K)? It is important

to the get these measurements for the two cases.

2) My impression is the that the resulting fitness landscape under inversion mutations is much smoother than that found under point mutations.

Please try to compare the outcome of the adaptation process in a scenario in which the ruggedness of the fitness landscapes under the two scenarios

are similar. Does this explain the pattern found in Figure 4?

3) What is the main reason underneath a no noticeable variation

of the average fitness with the mechanism when random neighbouring epistatic

interactions are assumed (up to K = 40)?

4) Is that possible to generalize the definition of roughness of the landscape along a given evolutionary trajectory, as

found in "Aita T, Iwakura M and Husimi Y 2001 A cross-section of the fitness landscape of dihydrofolate reductase

Protein Eng. 14 633"?

Reviewer #3: In this paper the authors study how structural variants, more specifically very small inversions, affect the topology of the fitness landscape and how this in turns affect the adaptive walks. This is a very relevant piece of information as most theory focus on single step mutations, and ignores the impact of larger types of mutations.

The results obtained suggest that by changing the connectivity between genotypes, inversions allow to cross fitness valleys and reach higher fitness peaks, reducing the number of times a walk ends up in a local fitness maxima.

I think this work is highly relevant and important and is an important step to introduce other type of mutations in fitness landscape theory. I really liked the idea behind the paper, but it was a bit complicated to follow sometimes, and so most of my comments focus on how the manuscript is written and I also have a few questions about the methods. I also have some minor comments that are at the end of the review.

Major comments

- The introduction could be shortened and go straighter to the point. Right now, it reads more as a historical introduction, and at the same time speaks very little of the work done (theoretical and empirical) on the impact of different types of mutations on the fitness landscape and the genotype to phenotype map (e.g. Aguilar-Rodríguez et al Evolution 2018, Alejandro V. Cano and Payne 2020 Plos Comp Biol, Zheng et al Science 2019 and see a review of some papers in Kemble et al 2019 Evolutionary applications). In addition, it would be good to explain why inversions are of interest. Maybe give some biological examples to contextualize the reader.

- The whole manuscript needs a revision to make it more focused, including the result section which is quite difficult to navigate. The theme at hand and the results obtained are not easy to explain, but although the subsections of the results make sense, there seems to be a lot of side thoughts throughout the results, which complicate readability.

- The first point of the results does not correspond to new results per se, so it should be in the methods section. I understand the need for the preamble, but it belongs in the methods section.

- I could not find the size of the inversions studied. This information should be easily accessible in the method description. In addition, it would be important to know the impact of the inversion size on the dynamics of the landscape.

- I do not understand the use of the conjugation to generate the inversion. Permutation should be enough to emulate the inversion since it corresponds to a change in the nucleotide order.

- The title does not reflect the part of the inversions, which is really the cool thing about this work. Maybe something like “Getting higher in rugged fitness landscapes with inversions” (not very inspired I know, but it is just an example).

The discussion is the most focused part of the manuscript, but I think it would be important to add two points:

- What is the impact for the accessibility and your conclusion of the sequence studied being circular vs linear?

- Is there any biological data that can support this type of result?

Finally, please revise your writing style as it overall seems too colloquial (I confess that I like it, because it shows the enthusiasm behind some of the results, which I share, but it is not the usually accepted one in journal articles).

Minor comments

Please revise/tone down the use of the term “Let us ...”

L2: Persuasive is a rather uncommon term for the fitness landscape metaphor, may be a more used one could visual or appellative.

L9: I like the informal style, but maybe use only note, instead of let’s notice.

L103: To sharpen the context  It is not clear what is meant here. Is it to contextualize?

Line 144: I find it weird the use of a scheme with numbering like an equation instead of a figure with both schemes, but I am not sure what are the policies of the journal regarding this.

L162: Table (capitalize the t).

L200: “Both graphs are not equivalent, although classical reasoning based on point mutations overshadows their differences!”  Please clarify this sentence.

L258: 259 - Since this is not always the case, maybe it would be better to add either: “Generally,” or “In most of the cases”. This way the reader knows that exceptions happen as is seen in figure 3.

L487 – schema  schematic no?

In the figures (to make the life easier to the reader) please put the name of the parameters (and then the symbols in parenthesis).

Table 1: please explain the meaning of each column name.

**Have the authors made all data and (if applicable) computational code underlying the findings in their manuscript fully available?**

Reviewer #1: Yes

Reviewer #2: Yes

Reviewer #3: Yes

PLOS authors have the option to publish the peer review history of their article (what does this mean?). If published, this will include your full peer review and any attached files.

Reviewer #1: No

Reviewer #2: No

Reviewer #3: No
---

## [Decision Letter · Decision Letter 1]

15 Sep 2022

Dear Dr. TRUJILLO,

Thank you very much for submitting your manuscript "Getting higher on rugged landscapes: Inversion mutations open access to fitter adaptive peaks in NK fitness landscapes" for consideration at PLOS Computational Biology. As with all papers reviewed by the journal, your manuscript was reviewed by members of the editorial board and by several independent reviewers. The reviewers appreciated the attention to an important topic. Based on the reviews, we are likely to accept this manuscript for publication, providing that you modify the manuscript according to the review recommendations.

The manuscript has been sent to the previous three reviewers. All of them generally support publication, with reviewers #1 and #3 pointing out some minor issues.

In addition, reviewer #1 also points out that the assumption of a binary alphabet may be quite crucial for the authors' conclusions. I would like to invite the authors to discuss this issue in their final version.

Sincerely,

Christian Hilbe

Academic Editor

PLOS Computational Biology

James O'Dwyer

Section Editor

PLOS Computational Biology

[LINK]

The manuscript has been sent to the previous three reviewers. All of them generally support publication, with reviewers #1 and #3 pointing out some minor issues.

In addition, reviewer #1 also points out that the assumption of a binary alphabet may be quite crucial for the authors' conclusions. I would like to invite the authors to discuss this issue in their final version.

Reviewer's Responses to Questions

**Comments to the Authors:**

Reviewer #1: The authors have addressed the majority of the comments from the reviewers. The manuscript is now much clearer.

After reading this manuscript, I realize there is one major underlying issue that was not raised in the first time. All of the theoretical and simulations work was done assuming a binary alphabet for the nucleotides. While this is a an extremely common and accepted approach, I believe it comes with an extra set of caveats here in this specific context (intragenic inversion, where the complementary sequence matters). Indeed, for the normal points mutations system, all genotypes that are separated by a hamming distance of 2, can be reached via 2 mutations, regardless of the size of the alphabet used. This no longer holds true for the inversion mutation system presented here for example AAAT and AAGG also have a Hamming distance of 2, yet they cannot be accessed with any amounts of inversion mutational steps (and inversion size) – something that a binary alphabet cannot capture. Based on this idea, under a 4 letter-alphabet, the better connectivity that fitness landscapes display through these intragenic mutations will be clearly reduced, with even some section being completely inaccessible from others. How this will affect the results described in the manuscript is however impossible to predict.

Minor comments:

At the beginning of the discussion (and conclusion), I believe it would be useful to remind the reader that the authors are talking about intragenic inversion. Because the phenomenon described here does not apply to larger inversions (to my understanding), which is what most people thing of, when mentioning inversions.

L 6-8: “Within this framework, the evolution of any population can be conceptualised as adaptive walks driven by successive mutations constrained by incremental fitness selection.” Evolution can also arise from random walks on a flat fitness plateau.

L42: I would be more precise here. Most models have focus on large inversions, where the main type of sequence disruption is caused by the break points themselves, which to rend the affected gene(s) non-functional. The kind of change the authors refer to here is unique to the extremely small inversions investigated here (and that’s why they are not included in the more general inversion models).

L55: “Recombination is not considered, so we are modelling asexual replication.” The causality in this sentence has been reversed. I would suggest editing it to “We are modelling asexual replication, therefore recombination is not considered.”

L304-305: I am not sure the use of the word “verify” here is correct. If the authors simply want to describe the figure, maybe the use of “observe” or “display” would be clearer. Alternatively, if the authors are checking some theoretical prediction against the simulations results, then a reference to said prediction would be really helpful.

Reviewer #2: The authors had satisfactorily addressed the points raised in my report. New material was added, and particularly, measures of roughness

of the fitness landscape upon inversions are now presented. In my opinion the manuscript is suitable for publication in its present form.

Reviewer #3: I really like this manuscript and I think the authors made a very good job in answering all reviewers comments and incorporating them in the manuscript. I have only small comments here and there.

The only problem was that in the clean version I was not able to see the figures and there was no legend for figure S1, but I am not sure if this was due to the formatting issue or not. All lines in the comments refer to the tracked version (where I could see the figures).

Minor comments

L190 - The reference to the figure with epistatic interactions (which I guess is figure S1) is missing

L228 - we want to verify if (instead of that)

L232 - Remove the Now in the beginning of the sentence

L248 - closets  closest :p

L284:285 - Maybe say it the other way around: Simulations with linear chromosomes show no significant difference from our reference circular model (see supplementary S1 Text, section 2 for detailed results).

L317: 318 - This a weird formulation for the sentence. Maybe something like: "Therefore, our results show that in the presence of inversions it is possible to reach higher fitness when compared to adaptive walks with only point mutations."

L320:321 - The beginning of the sentence is weird

L392, 430 - graph-theoretical ?

L428 - evolutive  evolutionary

L453: 455 - i don't understand the use of opposite in the sentence. And the following sentence is quite cryptic, what is the former and the latter? Please clarify

Legend of figure 3 - the reference for the figure with epistatic interactions is missing. Also it would be good to add a small sentence for people to note that the color scales differ in range between panels.

Legend of figure 4 - Instead of "The behaviour of the higher reached fitness vs the epistatic parameter K" maybe something like Changes in mean final fitness for different epistatic parameters (K) for ...

For the inset legend it would be better to add that this done for random and adjacent neighbouring epistatic interactions.

Supplementary text - Section 2.2 line 57 - linear inversions mainly affect -> do you mean the presence of inversions in linear chromosomes mainly affects large sized inversions?

Figure S1 needs a legend

**Have the authors made all data and (if applicable) computational code underlying the findings in their manuscript fully available?**

Reviewer #1: Yes

Reviewer #2: Yes

Reviewer #3: Yes

PLOS authors have the option to publish the peer review history of their article (what does this mean?). If published, this will include your full peer review and any attached files.

Reviewer #1: No

Reviewer #2: **Yes: **Paulo R. A. Campos

Reviewer #3: No

Figure Files:

Data Requirements:

Reproducibility:

References:

---

## [Editor Report · Decision Letter 2]

9 Oct 2022

Dear Dr. TRUJILLO,

We are pleased to inform you that your manuscript 'Getting higher on rugged landscapes: Inversion mutations open access to fitter adaptive peaks in NK fitness landscapes' has been provisionally accepted for publication in PLOS Computational Biology.

Best regards,

Christian Hilbe

Academic Editor

PLOS Computational Biology

James O'Dwyer

Section Editor

PLOS Computational Biology

I would like to thank the authors for taking into account all the reviewers' final comments (and I thank the reviewers for making all these constructive suggestions in the first place).

The manuscript has been further improved and is now ready for publication.

---

## [Editor Report · Acceptance letter]

21 Oct 2022

PCOMPBIOL-D-22-00459R2 

Getting higher on rugged landscapes: Inversion mutations open access to fitter adaptive peaks in NK fitness landscapes

Dear Dr Truillo,

I am pleased to inform you that your manuscript has been formally accepted for publication in PLOS Computational Biology. Your manuscript is now with our production department and you will be notified of the publication date in due course.

With kind regards,

Zsanett Szabo
